# Research on the Introduction of a Robotic Process Automation (RPA) System in Small Accounting Firms in Taiwan

**Hsing-Hua Hsiung * and Juo-Lien Wang**

Department of Accounting, Faculty of Management, Chaoyang University of Technology, 168 Jifeng E. Rd., Wufeng District, Taichung 41349, Taiwan
* Correspondence: sandyhsi@cyut.edu.tw

**Abstract:** This study explores the characteristics that influence the success factors of accounting firms in the introduction of a RPA system. The RPA system success factors used in this paper are based on the measurements of the Technology Acceptance Model (TAM) and the Information System Success Model (ISS). In this paper, a questionnaire survey method was used, and a total of 140 questionnaires were distributed to 70 small accounting firms in Taiwan. The results of this study showed that three characteristic factors—male, higher familiarity with the system and high CEO support—were significantly positively correlated with the success factors of the RPA system in accounting firms. Due to resource constraints, small- and medium-sized accounting firms have gone through a more difficult journey of digital transformation than the Big Four. Therefore, research on digital transformation with small- and medium-sized firms as samples is a topic worthy of attention. The research conclusions of this paper can provide a reference for the accounting digital transformation strategies of the accounting industry and educational institutions.

**Keywords:** accounting firms; Robotic Process Automation (RPA); Digital Automation System; Technology Acceptance Model; Information System Success Mode; digital transformation





## 1. Introduction

Digital transformation is a must for most businesses around the world. With increasing competition in the industry, globalization and the advent of new technologies, companies may lose their market in the battle for digital transformation if they have not considered how to integrate digital technologies to improve operational efficiency, enhance customer experience and adjust business models.

In recent years, due to the low birth rate and changes in the social structure, the accounting industry in Taiwan has faced management problems including manpower recruitment—specifically, the sharp drop in recruiters and the high turnover rate of employees. Scholars have tried to clarify the reasons for these difficulties and propose solutions. Lai (2018) and Wen (2002) conducted research on how accounting firms retain their talent. The research concluded that improving the welfare of employees and improving the working environment are the top priorities; in addition, Zhang and Prybutok (2003) pointed out that developing accounting firms industrialization is also a key factor in the survival of the firm, and it shows that improving the employees' sense of work achievement and satisfaction is also an important factor.

Japan experienced the dilemma of a declining birth rate in the 1990s. Wang (2019) studied the relevant systems in Japan and summarized the three strategic directions proposed by the government to assist enterprises to respond: (1) Utilization and cultivation of human resources; (2) equipment upgrade, technology and big data application; (3) process simplification, marketing and outsourcing. In addition, many studies also pointed out that in the face of manpower shortage among accounting firms, it is a clear guideline to develop digital technology and improve operational efficiency (Liu 2020).

In recent years, in response to the vigorous development of energy conservation, carbon reduction, digitalization and e-commerce and the implementation of the e-invoice policy, the Ministry of Finance of Taiwan launched a trial operation for B2B and B2C enterprises to issue e-invoices in 2000 and 2005, respectively. Since 2018, the Taiwan government has actively guided computer invoice business operators to import electronic invoices or switch to other types of unified invoices. The Taiwan government is also committed to establishing a safe and reliable transaction environment for electronic invoices. According to statistics, the number of electronic invoices in Taiwan is increasing year by year. Electronic invoices for consumer channels have grown from 4.4 billion in 2014 to 7.5 billion (in NTD) in 2019, increasing to 3.1 billion, an increase of 70%. The consumption amount increased from 1.7 trillion yuan in 2014 to 5.5 trillion yuan in 2019, a growth rate of 3.23 fold.

Most countries have not adopted Taiwan's unified invoice system. Taiwan's unified invoice, in accordance with the Business Tax Law, the Tax Collection Law, the Commercial Accounting Law and other relevant regulations, also has the functions of commercial transaction vouchers, accounting vouchers and taxation vouchers. The implementation of electronic invoices has accelerated the feasibility of accounting and tax informatization in Taiwan. Taiwan's accounting information software industry, in view of the popularity of electronic invoices is conducive to the implementation of digital accounting and taxation, when Taiwan's accounting and taxation process digital transaction environment has become mature, in order to help accounting firms to enhance value and competitiveness, they have proposed various accounting and tax digital information products, dedicated to assisting the electronic accounting forms, digital document approval and other related information systems to improve the manual operation process.

In Taiwan, there is a clear distinction between the business items of the Big Four accounting firms and the small firms. The business items of the small firms are mainly accounting and tax filing, and the main service clients are small- and medium-sized enterprises, while the business items of the Big Four are mainly projects that are relatively diverse and extensive, with accounting and tax processing only accounting for a small part of the business, since they mainly engaged in more professional business projects, such as auditing and consulting services. The main service customers are mostly large enterprises.

Taiwan's corporate organizational structure is dominated by small- and medium-sized enterprises. In 2020, the number of small- and medium-sized enterprises was 1,548,835, accounting for 98.93% of the total number of entrepreneurs. Due to the limited scale and resources, small- and medium-sized enterprises only have a relatively low budget to invest in complicated and professional accounting and taxation work, so they especially need the accounting processing and taxation services provided by small accounting firms. Therefore, in Taiwan, the importance of small accounting firms is extremely high.

In addition, due to the great difference in scale between the Big Four and the small- and medium-sized accounting firms, the applicable resources and systems are also completely different. In their research paper, scholars Ghosh and Lustgarten (2010) and Chen and Cheng (2008) conduct research on accounting firms according to their size. When faced with digital transformation, the Big Four have abundant resources and the ability to integrate to face this trend. However, it is more difficult for small accounting firms to recruit human resources. Digital transformation is the only way to face the shortage of human resources. Therefore, for small firms, on the contrary, the digital transformation capability of the company is an issue worthy of attention. Therefore, this paper takes small- and medium-sized accounting firms with inferior digital ability as the object, and conducts this research with profound practical significance.

In addition, since the TAM model was designed to explore the behavior model of users accepting the new information system, this model was very suitable for explaining the behavior of users accepting the new information system after the introduction of RPA information technology by small accounting firms, so this paper uses the TAM theoretical model, in order to analyze the factors that affect users' acceptance of RPA technology.

Therefore, this research uses the well-known information company Winton System Co., Ltd. (Taipei, Taiwan), which has been supported and adopted by nearly 5000 accounting firms in Taiwan, and uses the Robotic Process Automation (automatic digital) system launched by this company in December 2017 with innovative functions such as intelligent identification, cloud connection, and data analysis. This study analyzes accountants' perceptions of satisfaction with the benefits of automated digital systems.

The information success factors of the RPA system, based on the Technology Acceptance Model (TAM) and the Information System Success Model (DeLone and McLean 1992, 2003), are: information quality, system quality, system satisfaction, system use benefit, system use attitude and use willingness to use the system; this paper attempts to explore the relationship between the characteristic variable gender, daily use time of the system, and CEO support and the above six factors of information system success. It is expected that the research results can provide a reference for the information industry, the finance and taxation industry and educational institutions to carry out accounting digital transformation strategies.

## 2. Literature Review

### 2.1. The Digital Transformation of the Accounting Industry

Higgins (2021) accurately describes that in the business society, change is the only truly constant truth. Whether disrupting an unprecedented business or battling to adapt to a new operating model, professionals across all industries find themselves dealing with significant change. Additionally, many of the changes in recent years have been driven by emerging technologies. A professional accounting firm that is regarded as a trustworthy professional advisory service industry by enterprises cannot escape the wave of digital transformation.

The Digital Transformation Initiative (DTI) was launched by the World Economic Forum in 2015 to serve as the focal point for new opportunities and themes arising from the latest developments in the digitalization of business and society. The DTI has analyzed the impact of digital transformation on: automotive, consumer goods, electricity, health care, logistics and media, digital consumption, digital enterprise and societal implications. In 2016, the initiative was extended to cover seven additional industries, including professional services, the platform economy, and societal value and policy imperatives.

World Economic Forum (2017) proposed that disruptive technologies are fundamentally changing the economics of professional services. Four themes will be central to capturing digital value for the industry and wider society: (1) Business Model Transformation, (2) Intelligent Automation, (3) Digital Agility and (4) Talent Empowerment. In order to meet the digital transformation trends, being a professional service member, accountings firms need new strategies and operational capabilities for success in the new platform economy.

According to PwC's Global Artificial Intelligence Study (2017), it is estimated that by 2030, AI will bring in USD 15.7 trillion in revenue to the market, boosting global GDP growth by 14%, becoming the largest business opportunity in today's rapidly changing market. Among them, markets with the greatest contribution to AI development may be China and North America. The industries that will benefit the most will be retail, financial services and health care.

With the advancement of information technology, digitization has become the trend of modern accounting operations. Four major accounting firms, including PwC, EY, Deloitte, and KPMG, have already started to automate their work processes and have applied a large number of digital tools to save manpower and time. In Taiwan, most of the Big Four firms have introduced Robotic Process Automation (RPA) to help improve the efficiency and quality of audit work, use Analytical Process Automation (APA) to improve quality inspection and provide customers with high value-added information; at the same time, many of them have developed digital application tools and platforms, such as combining blockchain and IoT technologies, or using the "Financial Blockchain Correspondence

Service" of the Financial Regulatory Commission of the Ministry of Finance to assist in auditing services. The accounting firm uses digital thinking to reduce the time-consuming and laborious confirmation process, allowing clients to grasp the progress of the project and understand the trajectory of the audit in a timely manner.

As for small- and medium-sized accounting firms in Taiwan, the impact of digital technology and manpower shortages cannot be avoided. In the past, the manual operation method was not only time-consuming and labor-intensive, but also had a high error rate and could not provide real-time information. Therefore, some accounting firms have introduced automatic digital accounting systems to handle related operation procedures. This is a major shift in Taiwan's accounting industry facing technological changes and moving towards digital transformation.

Computerization has indeed replaced the repetitive work of accountants, and the workload is greatly reduced compared with traditional operations. However, for analysis work that requires professional judgment, accountants are more burdened on role-playing. At present, accounting firms have moved towards more professional fields. In-depth expansion of service areas, including tax formulation, monitoring budget implementation, financial asset management, arranging financing, reviewing investment projects, handling mergers and acquisitions and other businesses.

In the past, most institutional forecasts pointed to the death knell of the accounting industry due to the development of IT. However, until 2022, which is after the COVID-19 pandemic, as mentioned by Higgins (2021), most studies instead changed their attitudes, suggesting that accountants, as with other professionals, need to worry more about the issue of environmental adaptation when it comes to IT than about being replaced. There is no doubt that digital transformation has fundamentally changed the playing field. Big data has become a rich resource that, when harnessed, can produce huge competitive benefits. For the accounting industry poised to harness the potential of digital tools, this shift is a rare opportunity rather than a threat.

To sum up the views of scholars, under the leadership of the technological revolution, if accountants have the ability to use cloud, automation, artificial intelligence and blockchain, they will cooperate more closely with other business disciplines in the IT environment. With technical support in a collaborative environment, the accounting team will consist of dedicated accounting professionals and experts from other business areas. Companies need to rely on the accounting profession to drive innovation with financial data, build a more resilient and agile supply chain, and formulate business management plans to promote business growth while ensuring continuity. Tomorrow's accountants are more relevant, strategic and creative than ever before.

*2.2. The Technology Acceptance Model (TAM) and the Information System Success Model (ISS)*

Chen and Bao (2007) concluded that any information technology needs to be used in order to exert its effectiveness. Therefore, in the research on the effectiveness of information integration, the cause of "use" behavior has been widely valued by scholars. The Theory of Reasoned Action (TRA) constructed by Fishbein and Ajzen (1975) can be used to describe and anticipate the motives of human behavior. Continuing the TRA theory, the Technology Acceptance Model (TAM) is a behavioral model that explores users' acceptance of new information systems, and is a behavioral idea model developed by Davis (1989) and Davis et al. (1989) based on the theory of rational behavior, as shown in Figure 1. This model can be used to explain the user's behavior of accepting new information systems in information technology, and analyze various factors that affect users' acceptance of new technology.

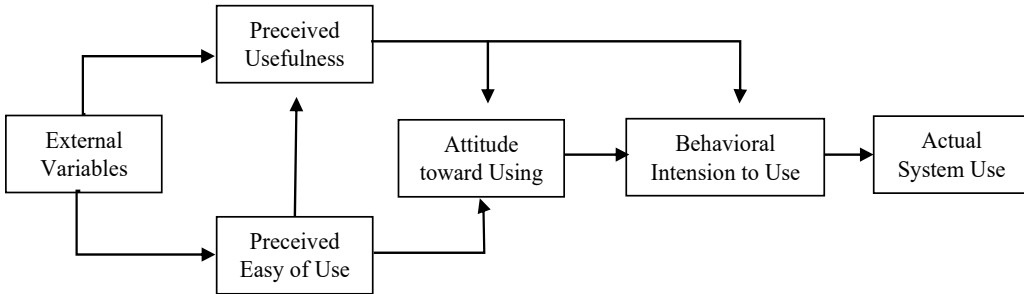

**Figure 1.** Technology Acceptance Model (Davis 1989).

The Technology Acceptance Model (TAM) believes that perceived usefulness and perceived ease of use are important prerequisites for the acceptance of information systems. The TAM has been applied to explain and predict user behaviors in accepting technological products; at the same time, it is also recognized that external variables can affect perceived usefulness, perceived ease of use, and intention to adopt the information system. So far, the TAM has been fully validated for various information systems, technical products, Internet activities e-commerce and e-learning related topics.

The Information System Success Model was proposed by (DeLone and McLean 1992). The model integrates previous research on e-commerce including communication research and the management information system, and builds a more consistent knowledge structure as a follow-up research on the effectiveness of information systems. The Information System Success Model was composed of six interrelated and causal dimensions, including system quality, information quality, system use, user satisfaction, individual impact, and organizational impact, as shown in Figure 2.

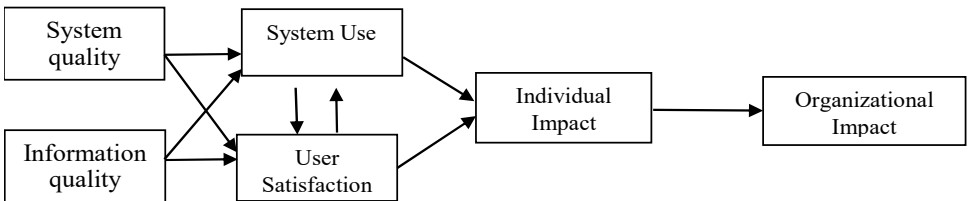

**Figure 2.** Information System Success Model (DeLone and McLean 1992).

In the original information system model in 1992, there is a close relationship between system use and user satisfaction. DeLone and McLean (2003) believe that use behavior will affect user satisfaction. When user satisfaction is higher, it will let users have higher use intentions, so use, use intentions, and user satisfaction generate a set of circular paths. In addition, both usage and user satisfaction will affect the net benefit of the information system. It is assumed that from the perspective of the owner or sponsor of the system, when the net benefit is user satisfaction, forms the second set of circular paths in this model, as shown in Figure 3.

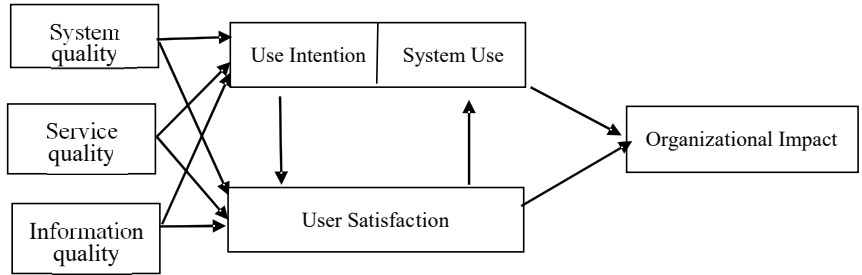

**Figure 3.** Modified Information System Success Model (DeLone and McLean 1992, 2003).

## 3. Research Design

### 3.1. Research Hypotheses

Legris et al. (2003) pointed out that the TAM model explores the reasons why users use information systems, but only has 40% explanatory power. Thus, while the TAM is a useful model, it must be integrated into a broader model to adequately address the factors behind the technology's acceptance, including variables related to human and societal change processes and the adoption of innovation models. For example, subjective factors such as support from senior managers, employee involvement, personal experience, education level, and job characteristics should also be incorporated into the analytical model.

This study divided the research hypotheses into three groups. The first group illustrates the effect of gender on the introduction of RPA systems. The second group is the CEO support, which will affect the introduction of the RPA system. The third group is the degree of system usage as a factor that affects the successful introduction of RPA systems.

*Group I: Gender and System Introduction*

According to the PwC's Report (2018), the article "Will robots really steal our jobs?" revealed that female workers could be more affected by automation over the next decade. Therefore, this study believes that it is imperative to understand the relevant research on information systems faced by accounting practitioners of different genders.

Zhang and Prybutok (2003) applied the Technology Acceptance Model (TAM) with gender as a moderating variable to understand the purchase intention of consumers' online shopping behavior. The conclusion of this study shows that gender is an important moderator variable affecting e-commerce behavior. The research results of behavior by Markauskaite (2006) also show that there is a significant difference between male and female gender and the ability to use IT. Additionally, it was found that technical application ability is beneficial to male workers and has obvious advantages. Teo et al. (2015) studied whether the results of the Technology Acceptance Model (TAM) differed by gender for pre-service teachers. The conclusion of the study pointed out that in professional teaching work related to IT, female teachers scored lower than male teachers in terms of attitude towards technology and intention to use technology. The findings of this study suggest that women face higher challenges in technology use than men. Based on the above, hypotheses H1~H6 of this research are formed:

**H1.** *Female workers are less satisfied with the information quality of the RPA system than male workers.*

**H2.** *Female workers are less satisfied with the system quality of the RPA system than male workers.*

**H3.** *Female workers experience less system satisfaction towards the RPA system than male workers.*

**H4.** *Female workers experience less system use benefits of the RPA system than male workers.*

**H5.** *Female system use attitude towards the RPA system is lower than male workers.*

**H6.** *Female willingness to use the RPA system is lower than that of male workers.*

*Group II: CEO Support and System introduction*

In addition, executive support is often cited as a key factor for organizations to take full advantage of IT. Jarvenpaa and Ives (1991) tested the CEO's behavior and perception in IT activities. Their analysis of industry manuals and annual letters to shareholders proves that although senior executives are not directly involved in IT activities, their supportive attitude towards IT is a key factor for organizations to fully utilize the advantages of IT. The research model designed by Liang et al. (2007) also provides evidence that top management can influence the usage of the enterprise resource planning (ERP) system by mediating the organizational-level pressure of the management system. The research model of Tallon et al. (2001) verifies that the IT goals and high-level management practices of enterprises

are the key determinants of the benefits for enterprises to practice IT. Therefore, this study forms hypotheses H7 to H12:

**H7.** *The extent to which the CEO supports the RPA system has a positive impact on information quality.*

**H8.** *The extent to which the CEO supports the RPA system has a positive effect on system quality.*

**H9.** *The extent to which the CEO supports the RPA system has a positive effect on system satisfaction.*

**H10.** *The extent to which the CEO supports the RPA system has a positive effect on system use benefits.*

**H11.** *The extent to which the CEO supports the RPA system has a positive effect on system use attitude.*

**H12.** *The extent to which the CEO supports the RPA system has a positive effect on willingness to use the system.*

*Group III: Degree of System Usage and System introduction*

DeLone and McLean (1992); Klenke and Kievit (1992) and other studies found that there is a high positive correlation between "use" and "user satisfaction". Siregar et al. (2017) stated in the paper that "user voluntary use" is an important factor affecting the acceptance of knowledge management technology. The above studies show that the higher the frequency of using the system, the more positive the attitude towards the information system. Therefore, this paper proposes that the longer the accountant uses the RPA system every day, the higher the perception of or satisfaction with the system. Therefore, this study forms hypotheses H13~H18:

**H13.** *Daily usage time of the RPA system positively affects information quality.*

**H14.** *Daily usage time of the RPA system positively affects system quality.*

**H15.** *Daily usage time of the RPA system positively affects system satisfaction.*

**H16.** *Daily usage time of the RPA system positively affects system use benefits.*

**H17.** *Daily usage time of the RPA system positively affects system use attitude.*

**H18.** *Daily usage time of the RPA system positively affects willingness to use the system.*

*3.2. Definition and Measurement of Research Variables*

This study adopts a closed questionnaire design to facilitate the statistics and quantification of data. The questionnaire is divided into three parts. The first part is the basic background information of the subjects, including gender, age, education, personal title and years of service; the area, the time of using the RPA system at work every day; CEO support for the system; the period of the company's introduction of the RPA system.

The third part of the questionnaire is based on the updated IS Success Model proposed by DeLone and McLean (2003). Additionally, they refer to the research variable structure of information success factors. The prototype of the research structure and the preliminary content of the questionnaire were developed. Additionally, through expert review, it was determined whether this design is consistent with the theme and scope of this research. Then, some unnecessary questions were deleted, confirming the research questions, and finally conducting a formal questionnaire.

The six factors of the Information System Success Model and items of this study, "information quality", "system quality", "system satisfaction", "system use benefits", "system use attitude" and "willingness to use the system", are shown in Table 1.

The total cost of measuring the error cost of the questionnaire and the relationship between the measuring cost and the total cost is relatively low when the five-point scale is used, so this study is designed with a five-point Likert scale.

Respondents answered based on their knowledge or satisfaction. The higher the level of agreement with the question item, the higher the rating score; and the lower the level of agreement with the RPA system, the lower the rating score.

**Table 1.** The measurements and the content of the questionnaire.

| Information Quality | System Use Benefits |
|---|---|
| 1. The system can provide the relevant information and reports I need. <br> 2. The information and reports provided by the system are complete, correct and reliable. <br> 3. The system can provide me with real-time information content. <br> 4. The system has a good layout and output format. | 1. Using the Digital Automation System can improve my work efficiency and performance. <br> 2. It saves my work time after using the Digital Automation System. <br> 3. The use of the Digital Automation System can effectively save manpower. <br> 4. After using the system, the service quality can be improved. |
| **System Quality** | **System Use Attitude** |
| 1. The system can improve work efficiency. <br> 2. The system is easy to operate and the process is smooth. <br> 3. The system can reduce the error rate. <br> 4. The system helps to keep the data intact. | 1. I think the Digital Automation System is helpful for me. <br> 2. I think the Digital Automation System is easy to use for me. <br> 3. I like to use the Digital Automation System. <br> 4. Using the Digital Automation System is a pleasant experience. |
| **System Satisfaction** | **Willingness to Use the System** |
| 1. I am satisfied with the functionality provided by the system. <br> 2. I am satisfied with the information content provided by the system. <br> 3. I think the Digital Automation System is a wise choice. <br> 4. After using Digital Automation System, its evaluation in my mind has improved. <br> 5. Overall, I am satisfied with the Digital Automation System. | 1. I would like to use the Digital Automation System. <br> 2. In the future, I would like to use the Digital Automation System. <br> 3. Overall, my willingness to use the Digital Automation System is quite high. <br> 4. I would recommend this system to others. |

### 3.3. Research Model

Regression analysis is a method of statistically analyzing data in order to understand whether two or more variables are correlated, the direction and strength of the correlation, and to build mathematical models that observe specific variables to predict variables of interest to researchers. In this study, the F test is used to determine whether the dependent variable and all independent variables have statistical significance; the $R^2$ is used to illustrate the explanatory power of the entire model; the t test of the beta coefficient is used to illustrate the influence of the independent variable on the dependent variable.

This study aims to understand whether gender, time to use the system every day, and CEO support for the RPA system influence the six success factors of the RPA system. We establish six regression models as follows:

$$\text{Model 1: MQ} = \beta_0 + \beta_1 \text{Sex} + \beta_2 \text{Time} + \beta_3 \text{Support} + \alpha_1 \text{Age} + \alpha_2 \text{Edu} + \alpha_3 \text{Scale} + \varepsilon \quad (1)$$

$$\text{Model 2: SQ} = \beta_0 + \beta_1 \text{Sex} + \beta_2 \text{Time} + \beta_3 \text{Support} + \alpha_1 \text{Age} + \alpha_2 \text{Edu} + \alpha_3 \text{Scale} + \varepsilon \quad (2)$$

$$\text{Model 3: SF} = \beta_0 + \beta_1 \text{Sex} + \beta_2 \text{Time} + \beta_3 \text{Support} + \alpha_1 \text{Age} + \alpha_2 \text{Edu} + \alpha_3 \text{Scale} + \varepsilon \quad (3)$$

$$\text{Model 4: SB} = \beta_0 + \beta_1 \text{Sex} + \beta_2 \text{Time} + \beta_3 \text{Support} + \alpha_1 \text{Age} + \alpha_2 \text{Edu} + \alpha_3 \text{Scale} + \varepsilon \quad (4)$$

$$\text{Model 5: AF} = \beta_0 + \beta_1 \text{Sex} + \beta_2 \text{Time} + \beta_3 \text{Support} + \alpha_1 \text{Age} + \alpha_2 \text{Edu} + \alpha_3 \text{Scale} + \varepsilon \quad (5)$$

$$\text{Model 6: AD} = \beta_0 + \beta_1\text{Sex} + \beta_2\text{Time} + \beta_3\text{Support} + \alpha_1\text{Age} + \alpha_2\text{Edu} + \alpha_3\text{Scale} + \varepsilon \quad (6)$$

The definitions of the above regression model independent variables are described as follows:

MQ: Information quality of the RPA system.
SQ: System quality of the RPA system.
SF: System satisfaction with the RPA system.
SB: System use benefits of the RPA system.
AF: System use attitude of the RPA system.
DF: Willingness to use the RPA system.

The definition of the dependent variable in the regression equation is described as follows:

Sex: The gender of the investigator.
Time: The time to use the voucher system at work every day.
Support: CEO support for the RPA system.

The definitions of control variables are described as follows:

Age: Respondent's age.
Edu: Educational level.
Scale: Firm scale.
$\varepsilon$: The residual term in the estimated regression equation.

## 4. Data Analysis and Results

### 4.1. Questionnaire Collection

Winton Sytem Co., Ltd. (Taichung, Taiwan) is the designated brand for the Taiwan software market. Established for 36 years, it has been supported by more than 25,000 users and is among the most widely used software companies by small- and medium-sized accounting firms in Taiwan. Therefore, this study surveyed respondents of accounting firms using an RPA system (automated digital system) developed by Winton Sytem Co., Ltd.

The questionnaire is an online questionnaire designed by the researchers. The purpose of the research is briefly explained in the questionnaire. The questionnaire survey was conducted in two stages. The first stage is to propose the prototype of the research structure and the preliminary content of the questionnaire from the literature review. The second stage is the distribution of the questionnaire. First, the questionnaire is distributed through the pre-test method, and 10 questionnaires are issued to experts and scholars to determine the design of the design—whether the questions can be consistent with the theme and scope of this research, and delete some unnecessary questions, confirm the research questions, and then start a formal questionnaire.

A total of 70 small- and medium-sized accounting firms were surveyed in this study, and each company distributed 2 questionnaires. The questionnaire is distributed and responded to by email. The survey period was from January 2020 to April 2020. In this study, a total of 140 questionnaires were distributed and 110 were recovered. Excluding 8 incomplete questionnaires (including some questions or missing basic information), there were 102 valid questionnaires, and the effective recovery rate was 72.85%.

### 4.2. Descriptive Statistical Analysis of Samples

#### 4.2.1. Sample Data

Among the 102 users, 62 were female, accounting for 60.8% of all valid samples. The total of 40 men accounted for 39.2% of all valid samples. The age distribution of 40 to 49 year olds accounted for the majority, a total of 28 people (27.5%), followed by under 25 year olds (22.5%). The educational background is dominated by university (including second and fourth skills), with a total of 54 people (accounting for 52.9%), followed by junior colleges (including second, third, and fifth colleges), with a total of 14 people (accounting for 13.7%).

In addition, the size of the firm is mostly less than 5 people, with 49 people (48%), followed by 5 to 15 people, with a total of 41 people (40.2%). The professional titles are mainly staff, with a total of 52 people. The number of years of service in the firm is less than 5 years, followed by 6–10 years and more than 20 years. The descriptive statistics related to the sample are shown in Table 2.

**Table 2.** Sample statistics.

| Items | | Frequency | % | Items | | Frequency | % |
|---|---|---|---|---|---|---|---|
| Gender | Male | 40 | 39.2% | During the company's introduction of the Digital Automation System | Unused | 23 | 22.50% |
| | Female | 62 | 60.8% | | Less than one year | 33 | 32.40% |
| Age | Under 25 | 23 | 22.50% | | 1~2 years | 32 | 31.40% |
| | 26~29 | 12 | 11.80% | | 3~4 years | 3 | 2.90% |
| | 30~39 | 17 | 16.70% | | 5+ years | 11 | 10.80% |
| | 40~49 | 28 | 27.50% | Use system time at work every day | Not full 1 h | 42 | 41.20% |
| | Over 50 | 22 | 21.60% | | 1~2 h | 27 | 26.50% |
| Education | High school | 14 | 13.70% | | 3~4 h | 10 | 9.80% |
| | College | 14 | 13.70% | | 5 h Above | 23 | 22.50% |
| | University | 54 | 52.90% | The area where the office is located | Northern Taiwan | 7 | 6.90% |
| | Graduate | 20 | 19.60% | | Central Taiwan | 91 | 89.20% |
| Job Title | Staff | 52 | 51.00% | | Southern Taiwan | 2 | 2.00% |
| | Director | 15 | 14.70% | | Outer islands of Taiwan | 2 | 2.00% |
| | Principal | 35 | 34.30% | Number of offices | 5 people or less | 49 | 48.00% |
| Years of Service | 5 years or less | 40 | 39.20% | | 5~15 people | 41 | 40.20% |
| | 6~10 years | 25 | 24.50% | | 16~30 peole | 5 | 4.90% |
| | 11~15 years | 7 | 6.90% | | More than 30 people | 7 | 6.90% |
| | 16~20 years | 6 | 5.90% | The level of support of the office director for the Digital Automation System | Low support | 19 | 18.60% |
| | 20+ years | 24 | 23.50% | | Medium support | 37 | 36.30% |
| | | | | | High support | 46 | 45.10% |

### 4.2.2. Descriptive Statistics for Digital Automation System Measurements

The descriptive statistics of the measurements are shown in Table 3 below. Overall, for the 102 users, in terms of information quality measurement, the degree of agreement was between 3.99 and 4.18—the subjects most agreed with "The system provides me with real-time information content" (4.18), and least agreed with "The system provides a good layout and output format" (3.99).

In terms of system quality measurement, the degree of agreement was between 3.85 and 4.48. Users most agreed with "The system helps to keep the data intact" (4.48), and least agreed with "The system can reduce the error rate" (3.85).

In terms of system satisfaction measurement, the degree of agreement was between 4.05 and 4.25—users most agreed with "My usage of the Digital Automation System is a wise choice" (4.25), and least agreed with "Overall, I am satisfied with Digital Automation System" (4.05).

Table 3. Data analysis table for measurements.

|  | Min | Max | Avg. | Std. |
|---|---|---|---|---|
| **Information quality** |  |  |  |  |
| The system provides the relevant information and reports I need. | 2 | 5 | 4.04 | 0.980 |
| The information and reports provided by the system are complete, correct and reliable. | 2 | 5 | 4.03 | 0.877 |
| The system provides me with real-time information content. | 2 | 5 | 4.18 | 0.844 |
| The system provides a good layout and output format. | 2 | 5 | 3.99 | 0.940 |
| **System quality** |  |  |  |  |
| The system can improve work efficiency. | 2 | 5 | 4.24 | 0.909 |
| The system is easy to operate and the process is smooth | 2 | 5 | 4.05 | 0.904 |
| The system can reduce the error rate. | 1 | 5 | 3.85 | 0.988 |
| The system helps to keep the data intact. | 2 | 5 | 4.48 | 0.714 |
| System satisfaction |  |  |  |  |
| I am satisfied with what the system offers. | 2 | 5 | 4.06 | 0.774 |
| I am satisfied with the information content provided by the system. | 2 | 5 | 4.10 | 0.744 |
| My usage of the Digital Automation System is a wise choice. | 2 | 5 | 4.25 | 0.824 |
| After using the Digital Automation System, it has improved in my mind. | 2 | 5 | 4.16 | 0.912 |
| Overall, I am satisfied with the Digital Automation System. | 2 | 5 | 4.05 | 0.890 |
| **System use benefits** |  |  |  |  |
| Using the Digital Automation System has improved my productivity and performance. | 1 | 5 | 4.16 | 0.823 |
| Save my work time after using Digital Automation System. | 1 | 5 | 4.13 | 0.911 |
| The use of the Digital Automation System can effectively save manpower. | 1 | 5 | 4.10 | 0.900 |
| After using the system, the service quality can be improved. | 1 | 5 | 4.04 | 0.869 |
| **System use attitude** |  |  |  |  |
| I think the Digital Automation System is helpful for me. | 1 | 5 | 4.19 | 0.878 |
| I think the Digital Automation System is easy to use for me. | 1 | 5 | 4.15 | 0.769 |
| I like to use the Digital Automation System. | 1 | 5 | 4.10 | 0.841 |
| Working with the Digital Automation System has been an enjoyable experience. | 1 | 5 | 4.01 | 0.899 |
| **Willingness to use the system** |  |  |  |  |
| I am willing to use Digital Automation System. | 1 | 5 | 4.25 | 0.940 |
| In the future, I hope to use the Digital Automation System. | 1 | 5 | 4.35 | 0.918 |
| Overall, my willingness to use the Digital Automation System is pretty high. | 1 | 5 | 4.34 | 0.946 |
| I would recommend this system to other | 1 | 5 | 4.11 | 1.031 |

In terms of system use benefits, the average degree of agreement was between 4.04 and 4.16—the subjects most agreed with "Using the Digital Automation System has improved my productivity and performance" (4.16), and least agreed with "After using the system, the service quality can be improved" (4.04).

In terms of system use attitude, the degree of agreement was between 4.01 and 4.19—the subjects most agreed with "I think the Digital Automation System is helpful for me" (4.19), and least agreed with "Working with the Digital Automation System has been an enjoyable experience" (4.01).

In terms of willingness to use the system, the degree of agreement was between 4.11 and 4.35—the subjects most agreed with "In the future, I hope to use the Digital Automation

System" (4.35), and least agreed with "I would recommend this Digital Automation System to Others" (4.11).

### 4.2.3. Reliability and Validity Testing

(1)  *Reliability Test*

From Table 4, we can see the six factors of the Information System Success Model in this study: "information quality", "system quality", "system satisfaction", "system use benefits", "system use attitude" and "willingness to use the system". The $\alpha$ coefficients of the questionnaires on the six measurements were between 0.900 and 0.964. The $\alpha$ coefficients of all facets are greater than 0.9, reaching a very credible level. Additionally, the overall reliability is higher than 0.984, indicating that the dimensions of this study have high consistency and stability whether individually or as a whole, so it has a considerable degree of reliability.

**Table 4.** Cronbach's $\alpha$ reliability analysis.

| Facet | Measure the Number of Questions | Cronbach's Coefficient |
| --- | --- | --- |
| Information quality | 4 | 0.925 |
| System quality | 4 | 0.900 |
| System satisfaction | 5 | 0.915 |
| System use benefits | 4 | 0.940 |
| System use attitude | 4 | 0.944 |
| Willingness to use the system | 4 | 0.964 |
| Total | 25 | 0.984 |

(2)  *Validity Test*

If the scale has reliability, it must have validity. Validity can generally be divided into three categories: content validity, construct validity and criterion validity (also known as criterion-related validity). This study only tested the content validity and construct validity of the measurement items.

Content validity refers to the appropriateness of the content of the scale, such as whether each item of the scale covers the characteristics of the variable to be measured, and whether the proportion of the number of items measuring each characteristic of the variable is appropriate. Since the content validity cannot be determined by any statistical test, if the content of the questionnaire comes from logical reasoning, a theoretical basis, experimental experience, and expert consensus, and has been pre-tested, it can be considered to have a considerable degree of content validity. After the first draft of the questionnaire for this study was established, various questions and doubts were raised according to the content of the questionnaire and revised. Then, the questionnaire pilot test was carried out. Finally, a total of 140 questionnaires were sent out. A total of 102 questionnaires were returned, and unnecessary questions were deleted before a complete questionnaire was drawn up, so it should have a certain degree of content validity.

Construct validity refers to the degree to which a measurement tool can measure the concept or characteristic of a theory. In order to confirm the construct validity of the scale, we generally need to test for convergent validity and discriminant validity. In terms of convergent validity, we used factor analysis to test the correlation of each item to other items. In this study, principal component analysis and maximum variation were used for factor extraction.

The Kaiser–Meyer–Olkin (KMO) statistic, which can vary from 0 to 1, indicates the degree to which each variable in a set is predicted without error by the other variables. A KMO value close to 1 indicates that the sum of partial correlations is not large relative to the sum of correlations and so factor analysis should yield distinct and reliable factors.

In this study, the obtained KMO values are 0.831 for information quality, 0.820 for system quality, 0.864 for system satisfaction, 0.793 for system use benefit, 0.850 for system

use attitude, and 0.851 for system use intention. The research questionnaire is suitable for factor analysis. The KMO statistic and the Bartlett test results are shown in Table 5.

**Table 5.** KMO and Bartlett test results.

| Factor Analysis Results | | Information Quality | System Quality | System Satisfaction | System Use Benefits | System Use Attitude | Willingness to Use the System |
|---|---|---|---|---|---|---|---|
| Kaiser–Meyer–Olkin | Sampling Suitability Quantity | 0.831 | 0.820 | 0.864 | 0.793 | 0.850 | 0.851 |
| Bartlett Sphere Test | Approximate chi-square distribution | 241.322 | 198.708 | 345.954 | 313.730 | 320.522 | 396.714 |
| | D.F | 6 | 6 | 10 | 6 | 6 | 6 |
| | Sig. | 0.000 | 0.000 | 0.000 | 0.000 | 0.000 | 0.000 |

In terms of discriminant validity, we use Pearson's correlation coefficient ($\gamma$) to detect whether there is no correlation coefficient greater than 0.8 or more than 0.9 between each factor in the correlation matrix. The results of the correlation analysis are shown in Table 6. The correlation coefficients for all factors were between 0.755 and 0.914. Discriminant validity was not demonstrated due to high construct correlation.

**Table 6.** Correlation analysis matrix.

| Items | Information Quality | System Quality | System Satisfaction | System Use Benefits | System Use Attitude | Willingness to Use the System |
|---|---|---|---|---|---|---|
| Information quality | 1.00 | | | | | |
| System quality | 0.858 ** | 1.00 | | | | |
| System satisfaction | 0.873 ** | 0.887 ** | 1.00 | | | |
| System use benefits | 0.820 ** | 0.851 ** | 0.914 ** | 1.00 | | |
| System use attitude | 0.767 ** | 0.832 ** | 0.866 ** | 0.865 ** | 1.00 | |
| Willingness to use the system | 0.755 ** | 0.852 ** | 0.890 ** | 0.914 ** | 0.902 ** | 1.00 |

Note: ** 5% significance level.

## 5. Results

This study aims to understand the characteristics that influence the success factors of accounting firms in the introduction of the RPA system by establishing six regression models and research hypotheses as well as conducting statistical tests. Table 7 shows the regression analysis results of this study.

**Table 7.** Regression analysis results.

| Dependent Var. | | MQ | | SQ | | SF | | SB | | AF | | DF | |
|---|---|---|---|---|---|---|---|---|---|---|---|---|---|
| | | COF. | *t* Value | COF. | *t* Value | COF. | *t* Value | COF. | *t* Value | COF. | *t* Value | COF. | *t* Value |
| Constant | | −1.447 | −1.925 | −1.544 | −1.996 ** | −0.877 | −1.071 | −0.59 | −0.712 | −0.549 | −0.744 | −0.442 | −0.525 |
| Independent Var. | Gender | −0.391 | −2.052 ** | −0.345 | −1.758 * | −0.436 | −2.099 ** | −0.512 | −2.435 ** | −0.707 | −3.779 *** | −0.558 | −2.614 ** |
| | Time | 0.201 | 2.394 *** | 0.227 | 2.627 ** | 0.281 | 3.068 *** | 0.273 | 2.948 ** | 0.212 | 2.57 ** | 0.168 | 1.785 * |
| | Support | 0.913 | 5.161 *** | 0.914 | 5.02 *** | 0.689 | 3.577 *** | 0.63 | 3.226 *** | 0.854 | 4.92 *** | 0.778 | 3.92 *** |
| Control | Age | −0.021 | −0.335 | −0.042 | −0.659 | −0.02 | −0.291 | −0.002 | −0.035 | −0.018 | −0.287 | −0.02 | −0.282 |
| | Edu | −0.228 | −2.016 ** | −0.088 | −0.757 | −0.145 | −1.18 | −0.184 | −1.476 | −0.146 | −1.313 | −0.173 | −1.36 |
| | Scale | 0.123 | 1.107 | −0.174 | −1.519 | −0.135 | −1.115 | −0.075 | −0.613 | −0.211 | −1.933 * | −0.179 | −1.44 |
| F | | 12.09 | | 10.749 | | 8.32 | | 7.818 *** | | 12.992 | | 7.241 | |
| R² | | 0.502 | | 0.472 | | 0.41 | | 0.395 | | 0.52 | | 0.376 | |
| Adj R² | | 0.461 | | 0.429 | | 0.36 | | 0.344 | | 0.48 | | 0.324 | |

Note: *** 1% significance level; ** 5% significance level; * 10% significance level.

Using Model 1 to test hypotheses H1, H7 and H13, the fit of this regression model was quite good (F value was 12.09 and *p* value was close to 0), and the overall explanatory power was 46.1%. The variables "gender", "daily usage time of the system" and "CEO support" were significantly positively correlated with information quality (the regression coefficients were −0.391, 0.201 and 0.913, respectively), verifying hypotheses H1, H7 and H13 proposed in this study. That is, "gender" was statistically significant, the coefficient direction value had a negative value, indicating that female cognitive satisfaction with "information quality" was lower than that of male. The extent to which the CEO supports the RPA system positively affects "information quality"; "daily usage time of the system" positively affects "information quality".

Using Model 2 to test hypotheses H2, H8 and H14, the fit of this regression model was quite good (F value was 10.749 and *p* value was close to 0), and the overall explanatory power was 42.9%. The independent variables "gender", "daily usage time of the system" and "CEO support" were significantly correlated with the "system quality" of the RPA system (the regression coefficients were −0.345, 0.227 and 0.914, respectively), verifying hypotheses H2, H8 and H14 proposed in this study. That is, "gender" was statistically significant, the coefficient direction also had a negative value, indicating that women's cognitive satisfaction with "system quality" was lower than that of men. The extent to which the CEO supports the RPA system positively affects satisfaction with "system quality". "Daily usage time of the system" positively affects satisfaction with "system quality".

In this study, Model 3 was used to test hypotheses H3, H9 and H15, the fit of this regression model was quite good (F value was 8.32 and *p* value was close to 0), and the overall explanatory power was 36%. The independent variables "gender", "daily usage time of the system" and "CEO support" were all significantly correlated with "willingness to use the RPA system" (the regression coefficients were −0.436, 0.281, 0.689, respectively), verifying the proposed hypotheses H3, H9 and H15. That is, women's awareness of "system satisfaction" of the RPA system was lower than that of men; the extent to which the CEO supports the RPA system positively affects the perception of "system satisfaction"; "daily usage time of the system" positively affects the perception of "system satisfaction".

Using Model 4 to test hypotheses H4, H10 and H16, the fit of this regression model was quite good (F value was 7.818 and *p* value was close to 0), and the overall explanatory power was 34.4%.

The independent variables were "gender", "daily usage time of the system" and "CEO support" were all significantly correlated with "system use benefits" (the regression coefficients were −0.512, 0.273, and 0.63, respectively). Hypotheses H4, H10 and H16 proposed in this study are fully supported. That is, women's perception of "system use benefits" of the RPA system was lower than that of men; the extent to which the CEO supports the RPA system positively affects the perception of "system use benefits"; "daily usage time of the system" positively affects awareness of "system use benefits".

Using Model 5 to test hypotheses H5, H11 and H17, the fit of this regression model was quite good (F value was 12.992 and *p* value was close to 0), and the overall explanatory power was 48%. The independent variables "gender", "daily usage time of the system" and "CEO support" were all significantly correlated with "willingness to use the system" (the regression coefficients were −0.707, 0.212, and 0.854, respectively). The statistical results also fully support hypotheses H5, H11 and H17 proposed in this study. That is, women's "willingness to use the system" was lower than that of men; the level of CEO support for the RPA system has a positive impact on "willingness to use the system"; "daily usage time of the system" has a positive impact on "willingness to use the system".

Finally, using Model 6 to test hypotheses H6, H12, and H18, the fit of this regression model was quite good (F value was 7.241 and *p* value was close to 0), and the overall explanatory power was 32.4%. The independent variables "gender", "daily usage time of the system" and "CEO support" were significantly correlated with "system satisfaction" (the regression coefficients were −0.558, 0.278 and 0.55, respectively). The results of this

study also support the proposed hypotheses H6, H12 and H18. That is, the independent variable "gender" was statistically significant, the coefficient direction had a negative value, indicating that women were less sensitive to "system satisfaction" than men. The extent to which the CEO supports the RPA system positively affects "system satisfaction"; "daily usage time of the system" positively affects "willingness to use the system".

In conclusion, the analysis of the regression model of this study concluded that all of the research hypotheses were valid.

## 6. Conclusions

This research explores the factors that affect the success of the RPA system used by small accounting firms in Taiwan. In addition to analyzing users' perceptions of the system, it also discusses accounting staff gender, CEO support and daily usage time of the system, and their impact on the RPA system. After reviewing the relevant literature on information technology, this research uses the Technology Acceptance Model (Davis 1989) and the Information System Success Model (DeLone and McLean 1992, 2003) to construct a model for evaluating the effectiveness of the system—information quality, system quality, system satisfaction, system use benefits, system use attitude and willingness to use the system—and established six regression models to verify the hypotheses.

This research is based on the RPA (Digital Automation) System launched in December 2017 by Wenzhong Information Company, a well-known information company adopted by most small- and medium-sized accounting firms in Taiwan, with innovative functions such as intelligent identification, cloud connection and data analysis. A total of 140 questionnaires were distributed to accountants who use the system, and 120 were recovered—excluding 8 incomplete questionnaires, 102 were valid questionnaires, and the effective recovery rate was 72.86%.

This study calculated the average value of each information success factor—information quality, system quality, system satisfaction, system use benefits, system use attitude and willingness to use the system. Based on the above, the respondents expressed positive comments on each measurement item, and the average score was greater than 3.85, indicating that users have a high degree of acceptance of and satisfaction with the RPA system in the accounting firm. In addition, the director of the firm also showed a high level of positive support for the RPA system.

According to the empirical results of the six regression models established in this study, "gender", "daily usage time of the system", and "CEO support" play a role in the electronic RPA system and the parameters that affect system performance. All of the estimated values are statistically significant, and the model fit is also good. This shows that the model and hypotheses proposed in this study are supported.

I.	*Managerial Implication*

The conclusion of this study is that "daily usage time of the system" is related to "information quality", "system quality", "system satisfaction", "system use benefits", "system use attitude" and "willingness to use the system". The information system benefit dimension was significantly positively correlated. This is consistent with the studies of DeLone and McLean (1992) and Klenke and Kievit (1992), which show that the degree of use of the system is an important factor that affects accountants' acceptance of the accounting RPA system.

Secondly, the firm's CEO support is also significantly positively correlated to the six information success factors. This is consistent with the studies of Jarvenpaa and Ives (1991), Liang et al. (2007), and Tallon et al. (2001), which also state that executive support is key for accountants to take full advantage of IT.

Finally, the variable "gender" is also significantly negatively correlated with all the six measurements of success information system model. Further, the coefficient values are all negative in the six models, which is consistent with Zhang and Prybutok (2003); Markauskaite (2006) and Teo et al. (2015) have reached the same conclusion, indicating that the challenges faced by women in the use of IT are indeed higher than those faced by men.

The managerial implications referring to action by/with leaders and the recommendations of this research are aimed at improving the acceptance of the digital system by accountants. It is necessary to strengthen CEO support for the system and increase use time, as these two factors can significantly and directly improve system success. In addition, it is also necessary to pay attention to the general phenomenon that female users are less accepting of new technologies than males, and on-the-job training can be strengthened.

The findings of this study show that a lack of information literacy or awareness among leaders is a significant barrier to digital transformation. Although enterprises generally have digital needs for informationization, digital tools must be introduced correctly and in real time through leaders. Additionally, a suitable implementation method must be planned to allow users to use more digital tools, introducing information-based benefits to enterprises to do more with less, so as to meet the fourth wave of the industrial revolution.

II.    *Practical/Social Implications*

Recently, the hot topic in technology is the application of new technologies such as artificial intelligence, the Internet of Things, big data, 5G and blockchain, and the digital transformation based on these technologies has received even more attention. Enterprises should not hesitate when it comes to digital transformation. There are many examples of companies losing market opportunities because they missed the transformation. For example, the mobile phone division of NOKIA was bought by Microsoft, which quit the mobile phone market because it missed the wave of smartphones; Blockbuster, a DVD rental company, went bankrupt because it missed the trend of streaming movies; it was also too late for the authors' to make digital cameras, and so that also ended in bankruptcy.

In addition, in recent years, there has been a severe shortage of labor in the accounting profession, which has led to a fierce competition for talent. However, raising salaries is not the only way. Through digital transformation, it is possible to reduce the repetitive and low value-added parts of the accounting firm's work. By improving job satisfaction and increasing the added value of the accounting firm's work, it is also an important thought direction.

In the face of digital transformation, the Big Four accounting firms have abundant resources and the ability to integrate to face this trend and change. However, it is more difficult for small accounting firms to recruit people, so the digital transformation capabilities of small firms are more worthy of attention.

Through the empirical results of this study, small accounting firms can successfully introduce the RPA system according to the framework of the Information System Success Model. The digital transformation benefits of the accounting service industry can not only improve the work efficiency and job satisfaction of employees to solve the problem of manpower shortage, but also improve the value-added process of small accounting firms, and use new tools and new ideas to devote themselves to process improvement to create new value in the accounting service industry.

III.    *Research Limitation*

This study mainly uses the Technology Acceptance Model and the Information System Success Model to measure the success of the accounting Digital Automation System. Due to scale of this research, this paper only includes some important variables, which is a limitation of this paper. In addition, the questionnaire survey method used in this study is also limited by an insufficient sample size. Suggestions for follow-up research include not only incorporating other variables of the Technology Acceptance Model or the Information System Success Model, but also incorporating on-site interviews and other methods to gain a deeper understanding of the use of RPA systems by accountants.

**Author Contributions:** Conceptualization, H.-H.H.; methodology, H.-H.H.; software, H.-H.H. and J.-L.W.; validation, H.-H.H. and J.-L.W.; formal analysis, H.-H.H.; investigation, H.-H.H.; resources, H.-H.H.; data curation, H.-H.H.; writing—original draft preparation, H.-H.H.; writing—review and

editing, J.-L.W.; visualization, H.-H.H.; supervision, J.-L.W.; project administration, H.-H.H. and J.-L.W. All authors have read and agreed to the published version of the manuscript.

**Funding:** This research received no external funding.

**Informed Consent Statement:** Informed consent was obtained from all subjects involved in the study.

**Conflicts of Interest:** The authors declare no conflict of interest.

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
