# Peer review of "Research on the Introduction of a Robotic Process Automation (RPA) System in Small Accounting Firms in Taiwan"

_economies, doi:10.3390/economies10080200_

Round 1

Reviewer 1 Report

Using a total of 140 questionnaires distributed to small accounting firms in Taiwan, current study shows that male, higher familiarity with the system, and high CEO’s support are positively correlated with the success factors of the RPA system in accounting firms. There are several comments to improve the readability of the study.

(1) This study is investigating the introduction of RPA and its effectiveness in small and medium-sized accounting firms in Taiwan. It would be good to present in more detail the reason for investigating small and medium-sized firms other than Big4 accounting firms. For example, the paper needs to explain whether RPA has already been introduced in Big4, whether RPA is more effective for small and medium-sized accounting firms, the number of accounting firms that have introduced RPA in Taiwan, and how RPA of Big4 is different from that of small and medium-sized accounting firms.

(2) It was suggested that a survey was conducted on small and medium-sized accounting firms in Taiwan, but the sample description below (shown in page 8) omitted which companies were targeted. It would be good to elaborate it.

“The questionnaire is an online questionnaire designed by the researchers. The purpose of the research is briefly explained in the questionnaire. It will be distributed and recycled by Internet mail. The survey period was from January 2020 to April 2020. In this study, a total of 140 questionnaires were distributed and 110 were recovered. Excluding 8 incomplete questionnaires (including some questions or missing basic information), there were 102 valid questionnaires, and the effective recovery rate was 72.85%.”

 (3) I am not sure whether the journal allows to use the full names of the authors when citing previous studies, but if not, authors might use the family name. For instance, Wang Junhong (2019) should be cited as Wang (2019). Please check journal’s policy and revise the citation format.

(4) More academic refinement is needed when writing the paper. In particular, check for typos or grammatical errors. For instance, in page 5, “Research hypothesie” is obviously wrong.

Author Response

Response 1: Thanks for the review's suggestion. We include the following in the Introduction section.(p.2)

In Taiwan, there is a clear distinction between the business items of the Big Four accounting firms and the small firms. The business items of the small firms are mainly accounting and tax filing, and the main service clients are small and medium-sized enterprises; while the business items of the Big Four are mainly The projects are relatively diverse and extensive, with accounting and tax processing only accounting for a small part of the business, mainly engaged in more professional business projects, such as: auditing and consulting services, etc. The main service customers are mostly large enterprises.

Taiwan's corporate organizational structure is dominated by small and medium-sized enterprises. In 2020, the number of small and medium-sized enterprises was 1,548,835, accounting for 98.93% of the total number of entrepreneurs. Due to the limited scale and resources, small and medium-sized enterprises only have a relatively low budget to invest in complicated and professional accounting and taxation work, so they especially need the accounting processing and taxation services provided by small accounting firms. Therefore, in Taiwan, the importance of small accounting firms is extremely high.

In addition, due to the great difference in scale between the Big Four and the small and medium accounting firms, the applicable resources and systems are also completely different. In their research paper, scholars Ghosh & Lustgarten (2006) Chen, Chang and Lee (2008) put the Accounting firms conduct research on topics according to their size. When faced with digital transformation, the Big Four have abundant resources and the ability to integrate to face this trend. However, it is more difficult for small accounting firms to recruit human resources. Digital transformation is the only way to face the shortage of human resources. Therefore, small firms On the contrary, the digital transformation capability of the company is an issue worthy of attention. Therefore, this paper takes the medium and small accounting firms with inferior digital ability as the object, and conducts this research with profound practical significance.

In addition, since the TAM model is designed to explore the behavior model of users accepting the new information system, this model is very suitable for explaining the behavior of users accepting the new information system after the introduction of RPA information technology by small accounting firms, so this paper uses the TAM theoretical model , in order to analyze the factors that affect users' acceptance of RPA technology.

Response 2: Thanks for the review's suggestion.We provide additional descriptions of the study sample.(p.9)

The questionnaire is an online questionnaire designed by the researchers. The purpose of the research is briefly explained in the questionnaire. The questionnaire survey was conducted in two stages. The first stage is to propose the prototype of the research structure and the preliminary content of the questionnaire from the literature review. The second stage is the distribution of the questionnaire. First, the questionnaire is distributed through the pre-test method, and 10 questionnaires are issued to experts and scholars to determine the design of the design. Whether the questions can be consistent with the theme and scope of this research, and delete some unnecessary questions, confirm the research questions, and then start a formal questionnaire.

A total of 70 small and medium accounting firms were surveyed in this study, and each company distributed 2 questionnaires. The questionnaire is distributed and responded to by email. The survey period was from January 2020 to April 2020. In this study, a total of 140 questionnaires were distributed and 110 were recovered. Excluding 8 incomplete questionnaires (including some questions or missing basic information), there were 102 valid questionnaires, and the effective recovery rate was 72.85%.

Response 3: Thanks for the review's suggestion.We have made changes to the citation descriptions.

Response 4: Thanks for the review's suggestion.We have revised the typos and grammar for the full text.

Reviewer 2 Report

I am pleased to have the opportunity to review this research paper. Although the topic of this research study is interesting and fits within the journal scope, I think authors should apply the comments indicated below to increase the quality of research justification, contributions, and findings. The manuscript lacks scientific style and structure.

Abstract: The aim and novelty of the research must be presented to understand why it is important this study. Motivated: Why Taiwan? Maybe you can also present a limitation of the study.

Introduction: Is a general description about the Taiwan system and the market, I think a better focus on the subject wear indicates.

The information success factors of the RPA system are based on the Technology Acceptance Model (TAM) and the Information System Success Model (DeLone and McLean 1992, 2003), which are: information quality, system quality, system satisfaction, system use benefit, system use attitude and use Willingness of the system; this paper attempts to explore the relationship between the characteristic variables: gender, daily use time of the system, and CEO’s support and the above six factors of information system successful. It is expected that the research results can provide reference for the information industry, finance and taxation industry and educational institutions to carry out accounting digital transformation strategies.

At the end of this paragraph maybe you can add some supplementary information about: What is the originality of this research?  Paper research gap and originality should be better presented at the end of the introduction section. Why Taiwan and why TAM and Information System Success Model

Literature Review

The digital transformation of the accounting industry

A better connection to the literature in the field. The authors connect it to different studies from Big 4 but we can find many studies in this area to see what is happening internationally and nationally so that we can identify the GAAP and the reason for the research.

Technology Acceptance Model (TAM) and Information Systems Success Model (ISS)

Methodology

Please link the methodology to the existing literature and extend the research hypothesis development. Maybe you can include the hypothesis also to the literature or to a separate part of the paper. For every Hypothesis I suggest a separate paragraph for the development of the hypothesis linked to the literature.

To motivate why five-point Likert Scale? Maybe a connection to other studies that also use the five-point Likert scale.

A detailed presentation of the pretesting of the questioner is needed to understand why it is important to pretest it.  

Research Model

Why regression? And how do the regression establish?

Descriptive Statistical Analysis of Samples

The α coefficients of all facets are greater than 0.9, reaching a very credible level. (Indicate the source)

KMO standard values or recommended values?

Results

Linked to other studies! What is different and what is common with the literature?

Conclusion:

Restructuration to:

-Managerial Implication

-Practical/Social Implications

To identify better the different implications of the study. So we can present what practical/professional and academic consequences will this study have for the future of scientific literature?

References

Add some new references in the field to connect the result of the paper to other studies to highlight the originality and novelty of the paper.

Good luck!

Author Response

Response 1:(Abstract). Thanks for the review suggestions. We've made changes to the abstract.

Due to resource constraints, small and medium-sized accounting firms have gone through a more difficult journey of digital transformation than the Big Four. Therefore, research on digital transformation with small and medium-sized firms as samples is a topic worthy of attention. The research conclusions of this paper can provide a reference for the accounting digital transformation strategies of the accounting industry and educational institutions.

Response 2:(Introduction).  Thanks for the review suggestions. We have added content for the introduction section.(p.2)

In Taiwan, there is a clear distinction between the business items of the Big Four accounting firms and the small firms. The business items of the small firms are mainly accounting and tax filing, and the main service clients are small and medium-sized enterprises; while the business items of the Big Four are mainly The projects are relatively diverse and extensive, with accounting and tax processing only accounting for a small part of the business, mainly engaged in more professional business projects, such as: auditing and consulting services, etc. The main service customers are mostly large enterprises.

Taiwan's corporate organizational structure is dominated by small and medium-sized enterprises. In 2020, the number of small and medium-sized enterprises was 1,548,835, accounting for 98.93% of the total number of entrepreneurs. Due to the limited scale and resources, small and medium-sized enterprises only have a relatively low budget to invest in complicated and professional accounting and taxation work, so they especially need the accounting processing and taxation services provided by small accounting firms. Therefore, in Taiwan, the importance of small accounting firms is extremely high.

In addition, due to the great difference in scale between the Big Four and the small and medium accounting firms, the applicable resources and systems are also completely different. In their research paper, scholars Ghosh & Lustgarten (2006) Chen, Chang and Lee (2008) put the Accounting firms conduct research on topics according to their size. When faced with digital transformation, the Big Four have abundant resources and the ability to integrate to face this trend. However, it is more difficult for small accounting firms to recruit human resources. Digital transformation is the only way to face the shortage of human resources. Therefore, small firms On the contrary, the digital transformation capability of the company is an issue worthy of attention. Therefore, this paper takes the medium and small accounting firms with inferior digital ability as the object, and conducts this research with profound practical significance.

In addition, since the TAM model is designed to explore the behavior model of users accepting the new information system, this model is very suitable for explaining the behavior of users accepting the new information system after the introduction of RPA information technology by small accounting firms, so this paper uses the TAM theoretical model , in order to analyze the factors that affect users' acceptance of RPA technology.

Response 3:(Literature Review).  Thanks for the review suggestions. This article has been corrected.

Response 4:(Methodology).  Thanks for the review suggestions. We have made modifications for the Research Design setion. 

(1)Research hypotheses have been written in sections.

(2)Explain why the questionnaire uses a five-point scale.

The total cost of measuring the error cost of the questionnaire and the relationship between the measuring cost and the total cost is relatively low when the five-point scale is used, so this study is designed with a five-point Likert Scale.

(3)Pre-test process description.

The questionnaire is an online questionnaire designed by the researchers. The purpose of the research is briefly explained in the questionnaire. The questionnaire survey was conducted in two stages. The first stage is to propose the prototype of the research structure and the preliminary content of the questionnaire from the literature review. The second stage is the distribution of the questionnaire. First, the questionnaire is distributed through the pre-test method, and 10 questionnaires are issued to experts and scholars to determine the design of the design. Whether the questions can be consistent with the theme and scope of this research, and delete some unnecessary questions, confirm the research questions, and then start a formal questionnaire.

A total of 70 small and medium accounting firms were surveyed in this study, and each company distributed 2 questionnaires. The questionnaire is distributed and responded to by email. The survey period was from January 2020 to April 2020. In this study, a total of 140 questionnaires were distributed and 110 were recovered. Excluding 8 incomplete questionnaires (including some questions or missing basic information), there were 102 valid questionnaires, and the effective recovery rate was 72.85%.

Response 5:(Research Model).  Thanks for the review suggestions. We have made modifications for the this parts.

(Describe why regression mode is used and how regression mode is established.

Regression Analysis is a method of statistically analyzing data in order to understand whether two or more variables are correlated, the direction and strength of the correlation, and to build mathematical models that observe specific variables to predict variables of interest to researchers. In this study, the F test is used to check whether the dependent variable and all independent variables have statistical significance; the R2 is used to illustrate the explanatory power of the entire model; the t test of the beta coefficient is used to illustrate the influence of the independent variable on the dependent variable.

Response 6:(Descriptive Statistical Analysis of Samples).  Thanks for the review suggestions. We have made modifications for the this parts. 

It can be observed through the Kaiser-Meyer-Olkin (KMO) statistic, which can vary from 0 to 1, indicates the degree to which each variable in a set is predicted without error by the other variables. A KMO value close to 1 indicates that the sum of partial correlations is not large relative to the sum of correlations and so factor analysis should yield distinct and reliable factors.

In this study, the obtained KMO values are 0.831 for information quality, 0.820 for system quality, 0.864 for system satisfaction, 0.793 for system use benefit, 0.850 for system used attitude, and 0.851 for system use intention. The research questionnaire is suitable for factor analysis. The results of KMO and Bartlett test are shown in Table 5.

Response 7:(Results).  Thanks for the review suggestions. We have made modifications for the this parts. 

  1. Practical/Social Implications

Recently, the hot topic of technology is the application of new technologies such as artificial intelligence, Internet of Things, big data, 5G and blockchain, and the digital transformation based on these technologies is even hotter. Enterprises should not hesitate about digital transformation. There are many examples of companies losing market opportunities because they missed the transformation. For example, the mobile phone division of NOKIA was bought by Microsoft and quit the mobile phone market because it missed the wave of smart phones; Blockbuster, a DVD rental company, went bankrupt because it missed the trend of streaming movies; It was too late to make up my mind to do my best to make digital cameras, and it also ended in bankruptcy.

In recent years, there has been a severe shortage of labor in the accounting profession, leading to a fierce competition for talent. The Big Four accounting firms are offering pay raises for their employees. Digital transformation is the only way to face manpower shortages. In the face of digital transformation, the Big Four have abundant resources and the ability to integrate to face this trend. However, it is more difficult for small accounting firms to recruit human resources. Therefore, the digital transformation capabilities of small firms are more worthy of attention.

This research hopes to use a practical research method to try to understand how small accounting firms that traditionally rely on human resources successfully introduce the automated process of RPA, so as to improve the value-added process of enterprise operations, improve efficiency, and reduce operating and labor costs. Through digital transformation The power to create new value for small and medium accounting firms.

Response 8:(References).  Thanks for the review suggestions. We have made modifications for the this parts. 

Round 2

Reviewer 2 Report

Pleas review table 3 because something is not ok (table description). Maybe because the layout.

Author Response

Response: Thanks for the reviewer's suggestion. We have made modifications for the description of Table 3.

4.2.2 Descriptive Statistics for Digital Automation System measurements

The descriptive statistics of the measurements are shown in Table 3 below. Overall 102 users, in terms of information quality measurement, the degree of agreement level is between 3.99 to 4.18, The subjects with the highest degree of user agreement is 「The system provides me with real-time information content」(4.18), the least agreement is 「The system provides a good layout and output format」(3.99).

In terms of system quality measurement, the degree of agreement level is between 3.85 and 4.48. Users are most agree with 「The system helps to keep the data intact」(4.48), and are least agree with 「The system can reduce the error rate」(3.85).

In terms of system satisfaction measurement, the agree level is between 4.05 and 4.25, users are most agreed with 「My usage the Digital Automation System is a wise choice.」(4.25), and the least agreed with 「Overall, I am satisfied with Digital Automation System」(4.05).

In terms of system used benefits, the average of agree level is between 4.04 to 4.16, most agree with 「Using the Digital Automation System has improved my productivity and performance」(4.16), least agreed with「After using the system, the service quality can be improved」(4.04).

In terms of system used attitude, the agree level is between 4.01 to 4.19, the most agreed item is 「I think Digital Automation System is helpful for me」( 4.19),the least agreed that 「Working with the Digital Automation System has been an enjoyable experience」(4.01).

In terms of the willingness to use the system, the degree of agree level is between 4.11 to 4.35, the most agreed item is: "In the future, I hope to use the Digital Automation System」(4.35), the least agreed item is 「I would recommend this Digital Automation System to Others」(4.11)".
